# Linking ecology and systematics of acidobacteria: Distinct habitat preferences of the *Acidobacteriia* and *Blastocatellia* in tundra soils

**Anastasia A. Ivanova[1], Alena D. Zhelezova[2], Timofey I. Chernov[2], Svetlana N. Dedysh[1] ***

**1** Winogradsky Institute of Microbiology, Research Center of Biotechnology of the Russian Academy of Sciences, Moscow, Russia, **2** Department of Soil Biology and Biochemistry, V.V. Dokuchaev Soil Science Institute, Moscow, Russia

\* dedysh@mail.ru

**Data Availability Statement:** The 16S rRNA gene sequence dataset used in this study is deposited with GenBank under the Bioproject accession number PRJNA497067 and is publicly available.

## Abstract

The *Acidobacteria* is one of the major bacterial phyla in soils and peatlands. The currently explored diversity within this phylum is assigned to 15 class-level units, five of which contain described members. The ecologically relevant traits of acidobacteria from different classes remain poorly understood. Here, we compared the patterns of acidobacterial diversity in sandy soils of tundra, along a gradient of increasing vegetation–unfixed aeolian sand, semi-fixed surfaces with mosses and lichens, and mature soil under fully developed plant cover. The *Acidobacteria*-affiliated 16S rRNA gene sequences retrieved from these soils comprised 11 to 33% of total bacterial reads and belonged mostly to members of the classes *Acidobacteriia* and *Blastocatellia*, which displayed opposite habitat preferences. The relative abundance of the *Blastocatellia* was maximal in unfixed sands and declined in soils of vegetated plots, showing positive correlation with soil pH and negative correlation with carbon and nitrogen availability. An opposite tendency was characteristic for the *Acidobacteriia*. Most *Blastocatellia*-affiliated reads belonged to as-yet-undescribed members of the family *Arenimicrobiaceae*, which appears to be characteristic for dry, depleted in organic matter soil habitats. The pool of *Acidobacteriia*-affiliated sequences, apart from *Acidobacteriaceae*- and *Bryobacteraceae*-related reads, had a large proportion of sequences from as-yet-undescribed families, which seem to specialize in degrading plant-derived organic matter. This analysis reveals sandy soils of tundra as a source of novel acidobacterial diversity and provides an insight into the ecological preferences of different taxonomic groups within this phylum.

## Introduction

The *Acidobacteria* is one of the most abundant and highly diverse bacterial phyla in soils and peatlands [1–6]. The proportion of *Acidobacteria*-affiliated 16S rRNA gene reads in sequence

**Funding:** This study was supported by the Russian Science Foundation (RSF -http://www.rscf.ru/en/ Field studies and 16S rRNA gene sequencing were performed by A.Z. and T.C. and supported by the project No 17-16-01057. Bioinformatic and taxonomic analyses were performed by A.I. and S. D. and were supported by the project No 16-14-10210. The funder had no role in study design, data collection and analysis, decision to publish or preparation of the manuscript.

**Competing interests:** The authors have declared that no competing interests exist.

pools retrieved from various soil habitats ranges between 5 and 50% of the total bacterial community [3,7–10]. Our knowledge of the roles of acidobacteria in soils includes decomposition of various biopolymers and participation in the global cycling of carbon, iron and hydrogen, but this list of functional capabilities remains far from being complete and is attributed to several sub-groups of this phylum only.

The currently explored diversity within the *Acidobacteria* is commonly addressed as corresponding to 26 major 16S rRNA gene sequence clades or subdivisions (SD) [11]. Recently, these 26 subdivisions were assigned to 15 class-level units, five of which contain described members [12]. These include three earlier established classes *Acidobacteriia*, *Blastocatellia* and *Holophagae* [13–15] as well as two recently proposed classes *Vicinamibacteria* and *Thermoanaerobaculia* [12]. The phylogenetic range of the class *Acidobacteriia* accommodates 16S rRNA gene sequences from several SDs including 1, 2, 3, 5, 11, 12, 13, 14, 15, and 24. The classes *Blastocatellia*, *Vicinamibacteria* and *Thermoanaerobaculia* correspond to one subdivision each, i.e. SDs 4, 6, and 23, while the class *Holophagae* includes SDs 8 and 22. In soil, acidobacterial SDs 1, 2, 3, 4 and 6 are the most abundant ones [8]. This is true for a wide range of ecosystem types, including boreal and tropical forests, grasslands and pastures, as well as arid landscapes [3–5, 8–10]. The abundances of subdivisions 1, 2 and 3 show negative correlation with soil pH, while the opposite tendency is characteristic for subdivisions 4 and 6 [8]. Peatlands are also among the preferred habitats of *Acidobacteria*. Acidic *Sphagnum*-dominated boreal peatlands are colonized mainly by members of SDs 1 and 3 [16–18]. Subarctic peatlands, in addition to SDs 1 and 3, may contain a relatively high proportion of SD2 *Acidobacteria* [19,20].

Despite their wide distribution in various soils, acidobacteria remain strongly underrepresented in culture collections due to difficulties in their cultivation and laboratory maintenance [6]. Much of the currently described diversity (ca. 15 genera) belong to the class *Acidobacteriia*, which includes the orders *Acidobacteriales* and *Bryobacterales* (SDs 1 and 3), and accommodates acidophilic or acidotolerant, mesophilic and psychrotolerant, chemoheterotrophic bacteria that utilize various sugars and polysaccharides and possess a number of hydrolytic capabilities [12]. The class *Blastocatellia* contains 7 genera of aerobic, mesophilic or thermophilic, chemo-heterotrophic bacteria that specialize on degradation of complex proteinaceous compounds and one genus of microaerophilic thermophilic anoxygenic photoheterotrophs, *Chloracidobacterium* [21]. Other classes of *Acidobacteria* contain only a limited number of characterized representatives and, therefore, their ecologically relevant traits remain poorly understood.

In the course of our recent study of prokaryotic community succession following sand fixation and soil formation in the tundra zone, *Acidobacteria* were identified as one of the most abundant bacterial groups, which was present along a whole gradient of increasing vegetation, from unfixed aeolian sands to mature soils under fully developed plant cover [22]. Since the phenotype of bacteria that colonize dry and nutrient-poor sands is clearly different from that of bacteria in organic-rich soils under plant cover, the acidobacterial populations in these sites with contrasting characteristics should have been quite distinct with regard to their lifestyles and environmental adaptations. The analysis presented here was undertaken in order to verify this hypothesis and to examine diversity patters of the *Acidobacteria* along a gradient of increasing vegetation in tundra.

## Materials and methods

### Sampling sites

The 16S rRNA gene sequence dataset retrieved by Zhelezova et al. [22] and deposited with GenBank under the Bioproject accession number PRJNA497067 was used for the analysis.

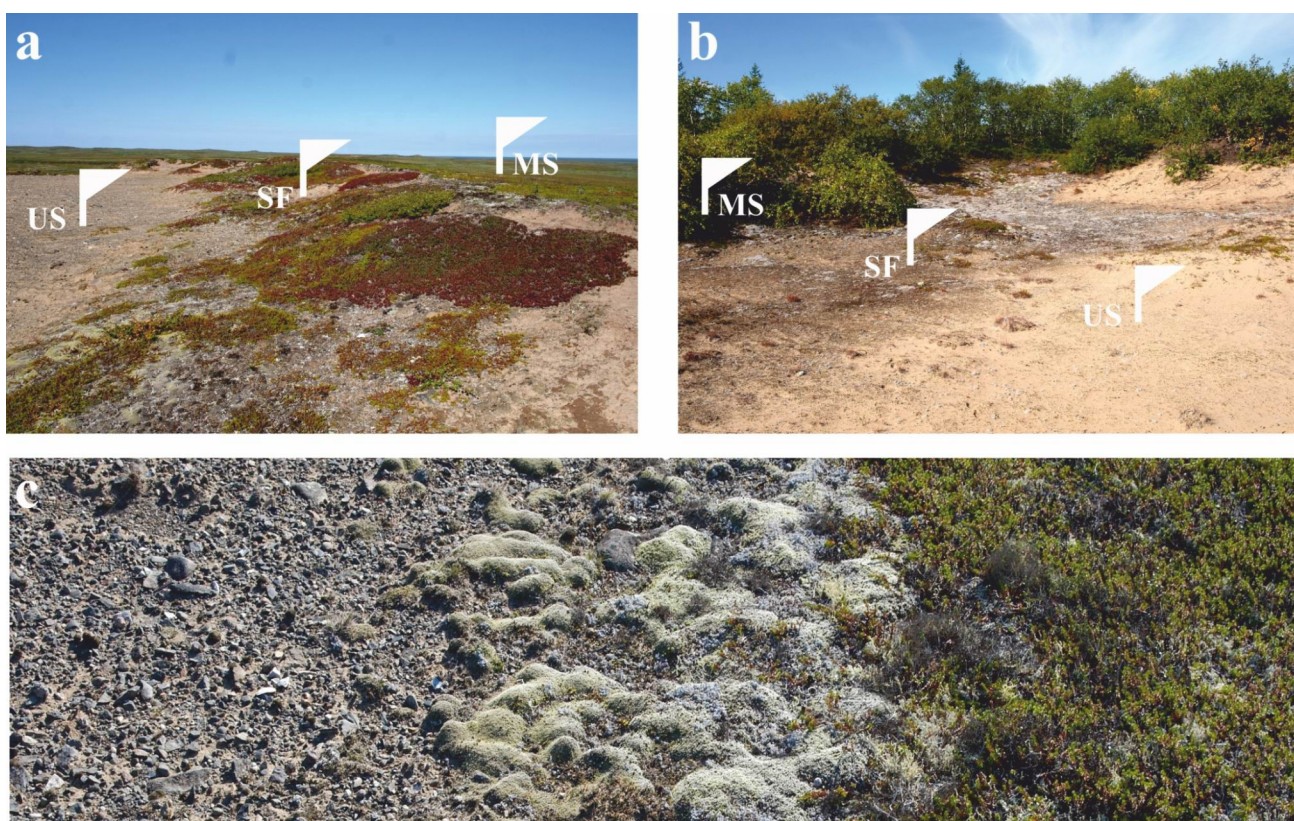

**Fig 1. Two sampling sites examined in the study.** (**a**) flat sand hills near Nelmin Nos (site I) and (**b**) small sand dunes near Naryan-Mar (site II). White flags indicate different types of plots along a gradient of increasing vegetation: US—unfixed sand, SF—semi-fixed surface, MS—mature soil. (**c**) Example of vegetation gradient from sand with gravel to cover of lichens and mosses (left to right) in site I near Nelmin Nos.

Two chronosequences of soil formation on aeolian sands with similar initial stages and different mature vegetation (typical tundra and wooded tundra) corresponded to the two sampling sites located on the shores of the Pechora River (Northwestern Russia, Nenetsia region). The two sites differed in the plant cover that developed on mature soil, i.e. typical tundra vegetation with subshrubs (Site I) and wooded tundra with rare trees and subshrubs (Site II). These sampling sites are shown in Fig 1 and the characteristics of soil samples used for the analysis are given in Table 1. Detailed description of the sampling sites are reported elsewhere [22]. No specific permits were required for sampling at these tundra sites. These locations are not privately-owned or protected in any way and are also not parts of national parks or reserves. Our sampling did not involve endangered or protected species.

**Table 1. Locations of sampling sites and characteristics of sampled substrates (values are shown as means (n = 2 for TOC and TN, n = 5 for pH) ± standard deviations).**

| Sampling sites | Surface type | Moisture, % | TOC, % | TN, % | pH |
|---|---|---|---|---|---|
| I–Nelmin Nos, 67˚58'34.3"N, 52˚55'19.9"E | US | 3.10 | 0.06 ± 0.00 | 0.03 ± 0.00 | 6.27 ± 0.27 |
| | SF | 2.61 | 0.15 ± 0.02 | 0.04 ±0.00 | 5.60 ± 0.29 |
| | MS | 27.86 | 1.71 ± 0.06 | 0.12 ± 0.01 | 5.37 ± 0.22 |
| II–Naryan-Mar, 67˚36'23.2"N, 53˚08'12.2"E | US | 0.39 | below detection limit | 0.02 ± 0.00 | 6.12 ± 0.10 |
| | SF | 0.82 | 0.18 ± 0.00 | 0.04 ± 0.00 | 5.66 ± 0.41 |
| | MS | 5.72 | 1.67 ± 0.26 | 0.09 ± 0.01 | 4.77 ± 0.25 |

For both sites, sampling was performed on three types of surfaces: 1- unfixed aeolian sand (US), 2- semi-fixed surface with mosses and lichens (SF), and 3—mature soil under developed plant cover (MS) (Fig 1, Table 1). Sampling plots of different types were located on a transect with 3–5 m between each plot. For every surface type on each site, five soil samples (~50 g of soil free of plants, mosses and lichens) were taken from depths of 1–5 cm. For molecular analyses, samples were stored at -70 ˚C. The total organic carbon (TOC) and total nitrogen (TN) contents were determined for the pooled sample of the five replicates taken for the molecular analysis from each plot using a Vario MACRO Cube CN-analyzer (Elementar Analysensysteme GmbH, Germany) [22]. pH was measured in soil suspension in distilled water in the ratio of 1:2.5.

### Analysis of SSU rRNA gene sequences

The sequences retrieved by Zhelezova et al. [22] were represented by the fragments of V3–V4 regions of bacterial and archaeal 16S rRNA genes. This pool of sequences was reanalyzed with *QIIME 2* v.2018.8 (https://qiime2.org) [23]. *DADA2* plugin was used for sequence quality control, merging of paired-end reads and chimera filtering [24]. Since the number of reads obtained from two replicates of MS plot in site I was too low (below 5.000 sequences), the comparative analysis of all three plots in site I was performed based on three replicates.

Operational Taxonomic Units (OTUs) were clustered applying VSEARCH plugin [25] with open-reference function using db Silva132 [26,27] with 97% identity. Taxonomy assignment was performed using BLAST against db Silva132 with 97% identity. 16S rRNA sequences affiliated with the *Acidobacteria* were extracted and used for further detailed analyses. The alpha-diversity indices were calculated using the core-metrics-phylogenetic method implemented in *QIIME 2* v.2018.8. Phylogenetic analysis was carried out using the ARB program package [28] (version 6.0.3) and the sequence alignment based on dbSilva132. Short sequences were added to the existing tree using the quick ARB parsimony insertion tool. The network diagram was constructed using Gephi [29].

### Statistical analyses

Statistical evaluations were made with GraphPad Prism (v. 7.0) applying multiple t-tests. Corrections for multiple comparisons were made using Holm-Sidak method. The significance level alpha was set at 0.05. Pearson correlation test was performed to check correlations between soil chemical properties and sequence abundances of different acidobacterial groups.

## Results

### Characteristics of sampled substrates along a gradient of increasing vegetation

The sampling plots corresponding to the three successional stages of soil formation in tundra —unfixed aeolian sand, semi-fixed surfaces with mosses and lichens, and mature soil under fully developed plant cover—differed from each other with regard to various physico-chemical parameters (Table 1). Most pronounced differences were observed with regard to moisture and total organic carbon contents. Unfixed and semi-fixed sands were extremely dry, while soils under plant cover contained more water. The content of organic carbon was nearly non-measurable in unfixed sands and increased gradually with formation of vegetation cover.

## Community composition of the *Acidobacteria*

A total of 1,019,619 partial 16S rRNA gene sequences (mean amplicon length 260 bp) were retrieved from the examined soil samples (Table 2). Of these, 463,507 sequences were retained after merging of paired-end reads, quality filtering, removing chimeras and singletons. The pool of *Acidobacteria*-affiliated reads included 74037 sequences, which accounted for 9–31% of all bacterial reads in different samples. Overall, the acidobacterial alpha-diversity in unfixed aeolian sands was slightly lower than that in soils of vegetated plots (see mean values of Shannon indexes in Table 2) but this difference was not statistically significant.

As revealed by principle coordinate analyses, the acidobacterial communities corresponding to the three successional stages of soil formation in tundra were also clearly distinct (Fig 2).

The *Acidobacteria*-affiliated sequences retrieved from the two tundra sites belonged to members of the classes *Acidobacteriia* (53–100% of all acidobacterial reads in different samples), *Blastocatellia* (0–34%), and *Holophagae* (0–13%) (Fig 3). The proportion of sequences that could not be assigned to these classes was in the range of 0–6% in unfixed sands and 0–1% in mature soils under plant cover. The pool of sequences from the *Acidobacteriia* was composed of reads from representatives of the orders *Acidobacteriales* (SD1) and *Bryobacterales* (SD3) as well as SD2 acidobacteria, which comprise an as-yet-undescribed order of this class. Notably, a large proportion of *Acidobacteriales*-affiliated sequences (6–37% of all

**Table 2. Sequencing statistics and various alpha-diversity metrics.**

| Sampling site | Sample ID | Raw reads | Filtered reads* | *Acidobacteria* reads | *Acidobacteria*/Filtered reads (%) | Diversity indices | | |
|---|---|---|---|---|---|---|---|---|
| | | | | | | Shannon | Observed OTUs | Pielou's evenness |
| SI (Nelming Nos) | US | 26888 | 11429 | 2392 | 21 | 4.12 ±0.09 | 24 | 0.88 |
| | | 48969 | 21871 | 4509 | 21 | | 29 | 0.85 |
| | | 39219 | 19133 | 3658 | 19 | | 31 | 0.84 |
| | SF | 43054 | 21171 | 3302 | 16 | 4.81 ±0.16 | 40 | 0.93 |
| | | 34967 | 16873 | 3026 | 18 | | 37 | 0.89 |
| | | 36727 | 17525 | 4368 | 25 | | 44 | 0.89 |
| | MS | 40752 | 17570 | 3991 | 23 | 4.57 ±0.03 | 40 | 0.92 |
| | | 28471 | 11381 | 1910 | 17 | | 24 | 0.92 |
| | | 30914 | 12816 | 3168 | 25 | | 37 | 0.89 |
| SII (Naryan-Mar) | US | 35900 | 17014 | 1585 | 9 | 4.08 ±0.26 | 26 | 0.87 |
| | | 43734 | 22605 | 2278 | 10 | | 31 | 0.85 |
| | | 83916 | 36714 | 3559 | 10 | | 30 | 0.89 |
| | | 37253 | 16094 | 1599 | 10 | | 19 | 0.86 |
| | | 42662 | 15079 | 1615 | 11 | | 26 | 0.86 |
| | SF | 51797 | 22265 | 2963 | 13 | 4.34 ±0.07 | 24 | 0.95 |
| | | 52038 | 23126 | 3344 | 14 | | 30 | 0.86 |
| | | 51165 | 23019 | 3379 | 15 | | 32 | 0.86 |
| | | 59837 | 26503 | 3848 | 15 | | 34 | 0.86 |
| | | 66002 | 29843 | 3221 | 11 | | 29 | 0.91 |
| | MS | 45105 | 24362 | 3721 | 15 | 4.44 ±0.65 | 40 | 0.92 |
| | | 36054 | 16322 | 3039 | 19 | | 23 | 0.87 |
| | | 22707 | 11672 | 3606 | 31 | | 59 | 0.91 |
| | | 32450 | 14527 | 3218 | 22 | | 23 | 0.88 |
| | | 29109 | 14593 | 2738 | 19 | | 24 | 0.88 |

*Filtered reads: number of merged paired-end sequences excluding low quality reads, singletons and chimeras.

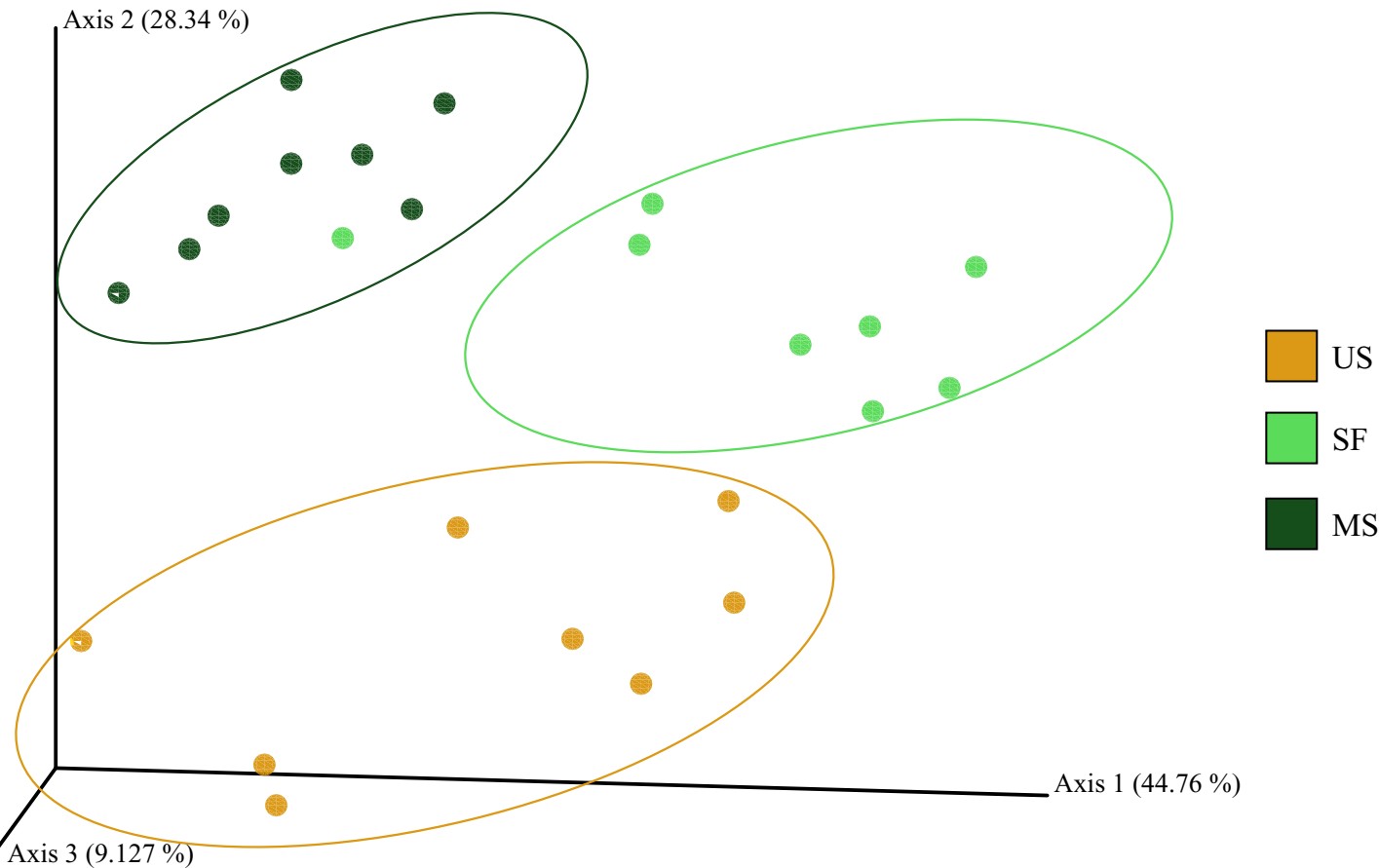

**Fig 2. Comparison of the *Acidobacteria* community composition in samples examined in this study by principle coordinate analyses (PCoA).** PCoA plot is based on the weighted UniFrac distance of the sequencing dataset.

acidobacterial reads in different samples) could not be assigned to the only currently described family of this order, i.e. *Acidobacteriaceae*. This group of sequences is further addressed as 'uncultured *Acidobacteriales*' in accordance with the taxonomic classification used in db Silva132 and indicated by black hatching in Fig 3.

Taxonomy-based analysis revealed a clear shift in the *Acidobacteria* community composition along a gradient of increasing vegetation in both study sites (Fig 3). The most characteristic feature of acidobacterial communities in unfixed sands was a prominent presence of the *Blastocatellia* (SD4), which comprised 31±2% and 14±5% of all *Acidobacteria*-affiliated reads obtained from sites I and II, respectively (S1 Table). The relative abundance of these bacteria declined dramatically in vegetated soils of both sites, down to 7±4% of reads in mature soils from the site I and a complete disappearance in mature soils from site II. The same trend was observed for members of the *Holophagae*, which accounted for 12±1% and 7±3% of all acidobacterial reads in unfixed sands from sites I and II, respectively, but were not detected in mature soils (Fig 3). The opposite trend was detected for members of the *Acidobacteriaceae* and SD2, which were either absent (*Acidobacteriaceae* in site I) or present in a low abundance in unfixed sands (13±5% for *Acidobacteriaceae* in site I and 4–7% for SD2 in both sites) but became major community members (with relative abundances of approximately 20–30%) in vegetated mature soils (S1 Table). The *Bryobacterales*-affiliated acidobacteria (SD3) displayed highest relative abundances in semi-fixed soils, accounting for 36±9% and 50±4% of all

acidobacterial reads in sites I and II, respectively. No clear trend could be observed for the uncultured group within the order *Acidobacteriales* (Fig 3).

## Most abundant OTUs of *Acidobacteria* and their distribution patterns

Using 97% sequence identity, a total of 232 acidobacterial OTUs were identified in all studied samples. Of these, 157 OTUs were detected in the site I and 159 OTUs were identified in the site II (Table 2); 80 OTUs were shared between sites I and II (S1 Fig). The numbers of OTUs shared between the two sites within each of the succession stages were 23, 24 and 42 for US, SF and MS plots, respectively (S1 Fig).

Our further, detailed analysis was focused on five major taxonomic groups within the *Acidobacteria*. Two of these groups, *Blastocatellia* (SD4) and *Acidobacteriaceae* (SD1), were chosen because they displayed two clearly opposite trends along a gradient of increasing vegetation in tundra soils. These groups of *Acidobacteria* contain a number of cultured and characterized representatives, thus offering a possibility of analyzing specific reasons behind their environmental distribution. Two other groups, SD2 and uncultured *Acidobacteriales*, do not contain described representatives and were chosen, therefore, in order to find out their eco-niche preferences. One additional group that was included in the analysis, *Holophagae* (SD8), contains only few cultured representatives; the physiology and functional capabilities of these acidobacteria remain poorly understood. The OTUs of these bacteria comprising ≥1% of all *Acidobacteria*-affiliated reads in at least one of the examined sites are listed in Table 3 and are displayed in S2 Fig.

Although the same taxonomic representation of *Acidobacteria* at the order and family levels was observed for both sites, diversity of OTUs identified in the sites I and II was clearly different (S2 Fig). For example, *Blastocatellia*-affiliated OTUs 1–3, which were among the most abundant OTUs in site I, were absent from site II. Vice versa, the most abundant *Blastocatellia*-affiliated OTU in site II, OTU 7, was not detected in site I. Other groups of *Acidobacteria* in the two sites were also represented by different OTUs, which may be due to some differences between the sites I and II, such as the plant cover composition and the moisture level (Table 1).

The most abundant OTUs of the *Blastocatellia* (OTUs 1–3 and 7–9) belonged to the as-yet-uncultivated clade RB41 within the family *Pyrinomonadaceae* and occurred mainly in unfixed sands in both tundra sites. Only three OTUs were classified at the genus level as representing the genera *Blastocatella* (OTUs 11 and 12) and *Stenotrophobacter* (OTU 5).

By contrast, nearly all of the *Acidobacteriaceae*-affiliated OTUs could be classified at the genus level and represented the genera *Granulicella* (OTUs 24–28, 30, 33, 34), *Acidipila* (OTUs 23, 32) and *Bryocella* (OTU 29). The most abundant OTU from the *Acidobacteriaceae* (No 31), however, could not be assigned to any of the described genera and displayed 96.0–96.5% similarity with 16S rRNA gene sequences from '*Acidisarcina*'/*Acidipila* group.

The most abundant SD2-affiliated OTUs (No 13–15) displayed high similarity with environmental 16S rRNA gene sequences retrieved from soils or sediments of various northern locations (GenBank numbers FJ004757, KM200371, FJ625349). These were detected mainly in mature soils, although OTU 13 was also highly abundant in semi-fixed sand from site I.

Finally, OTUs comprising the group of uncultured *Acidobacteriales* were present in all examined samples but varied with regard to their identity and relative abundance. Thus, OTUs 38 and 48 were abundant in unfixed sands but were not detected in semi-fixed sands and soils under plant cover. By contrast, OTU 39 was highly representative in semi-fixed sands but was also present in other samples. Environmental 16S rRNA gene sequences representing this group of acidobacteria were retrieved mainly from various grasslands and forest soils but

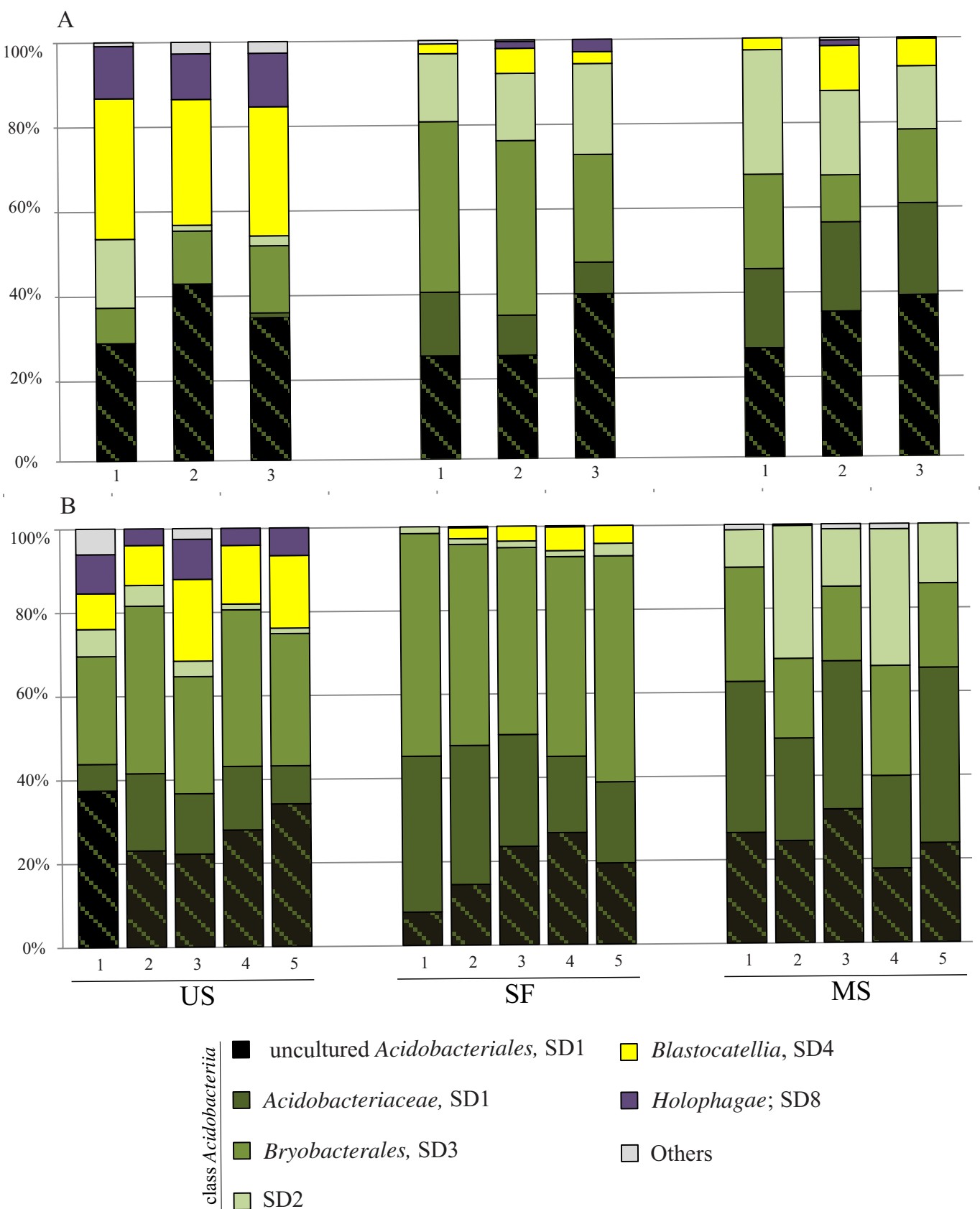

**Fig 3. Community composition of the *Acidobacteria* along a gradient of increasing vegetation–unfixed aeolian sand, semi-fixed surfaces with mosses and lichens, and mature soil under fully developed plant cover—based on 16S rRNA gene sequence analysis.** A–site I, B–site II. The taxonomic analysis was performed according to dbSilva 132. The values of the relative abundance of different acidobacterial groups in individual samples are given in S1 Table. Significant differences in relative abundances of particular acidobacterial groups between unfixed sand and different stages of soil formation were revealed for the *Acidobacteriaceae* (*P*-value < 0.01), *Bryobacterales* (*P*-value < 0.001) and *Blastocatellia* (*P*-value < 0.001) (S2 Table).

also from microbial mats of lava tube walls, orthoquartzite caves, uranium and potassium mines (Table 3).

Major trends of changes in the *Acidobacteria* community composition in sandy soils of tundra along the gradient of increasing vegetation are seen in the network diagram illustrating the most abundant OTU distribution between the three examined types of plots (Fig 4). An apparent shift from the community composed of the *Blastocatellia* and uncultured *Acidobacterales* in aeolian sands to the community dominated by various subgroups of the *Acidobacteriia* in mature soils is observed in this diagram.

### Specific phylogenetic sub-groups of *Acidobacteria* revealed in tundra soils

A large group of OTUs affiliated with the class *Blastocatellia*, which were identified mostly in unfixed sands (OTUs 1–4, 6–10), were classified as belonging to the as-yet-uncultivated clade RB41 within the family *Pyrinomonadaceae* (Table 3). This group was named after the environmental clone sequence RB41 (GenBank accession No Z95722), which was retrieved from the Roggenstein field site soil near Munich (Germany) in the seminal work of Ludwig et al. [1]. Detailed phylogenetic analysis of these sequences revealed that they form a distinct phylogenetic group, which clusters separately from the other described families of the order *Blastocatellales*, i.e. *Blastocatellaceae*, *Arenimicrobiaceae* and *Pyrinomonadaceae* (Fig 5). These sequences displayed 91–92% similarity to 16S rRNA gene sequences from members of the genera *Brevitalea* and *Arenimicrobium*, while their sequence similarity to other described members of the *Blastocatellia* was in the range of 86–89%.

Another large group of numerically abundant OTUs identified in tundra soils affiliated with the order *Acidobacteriales* but did not belong to the only currently described family *Acidobacteriaceae* (Fig 6). In db Silva 132, this group is addressed as 'uncultured *Acidobacteriales*', which is not fully correct since it comprises a number of isolates named "Ellin" (Ellin7137, 7522, 6547, 6528, 6527, 5106, 323 and others). These isolates were obtained by Janssen and co-workers [30–33]; none of them, however, was characterized, so that no information about these bacteria is currently available. 16S rRNA gene sequence similarity of these bacteria to described members of the family *Acidobacteriaceae* is below 91%.

### Correlations between chemical soil properties and the abundances of different acidobacterial groups

To reveal the influence of several chemical soil properties (pH, C and N contents) on the relative abundances of different acidobacterial groups, Pearson correlation test was performed. The positive correlation was revealed between pH and the relative abundances of the *Holophagae* (SD8) (R = 0.82, p = 0.05) and *Blastocatellia* (SD4) (R = 0.80, p = 0.06). The highest relative abundance of these groups was detected in the acidobacterial communities of unfixed sands in both sites. The negative correlation was revealed between pH and the relative abundances of the *Acidobacteriaceae* (SD1) (R = -0.83, p = 0.04) and SD2 (R = -0.73, p = 0.1), which tended to dominate in mature soils. These correlations could be explained by the gradual decrease of pH during the process of soil formation. No significant correlations, however, were revealed between C and N contents and the abundance of particular acidobacterial groups (Table 4).

**Table 3. The most abundant operational taxonomic units (OTUs) of *Acidobacteria* detected in tundra soils.**

| № OTU | Silvadb match | Taxonomy | Reported habitat | Similarity (%) |
|---|---|---|---|---|
| 1 | EU132342 | SD4, *Pyrinomonadaceae* RB41 | soil from an undisturbed mixed grass prairie preserve USA | 97.7 |
| 2 | EF019176 | SD4, *Pyrinomonadaceae* RB41 | trembling aspen rhizosphere USA | 98.0 |
| 3 | HQ645212 | SD4, *Pyrinomonadaceae* RB41 | soil samples from meadow in the Tibet Plateau China | 98.4 |
| 4 | Z95722 | SD4, *Pyrinomonadaceae* RB41 | soil sample Germany | 99.2 |
| 5 | EU132039 | SD4, *Stenotrophobacter* | soil from an undisturbed mixed grass prairie preserve USA | 96.9 |
| 6 | EF494321 | SD4, *Pyrinomonadaceae* RB41 | River granitic landscape Australia | 97.3 |
| 7 | EU150230 | SD4, *Pyrinomonadaceae* RB41 | dry meadow soil USA | 97.3 |
| 8 | JN615840 | SD4, *Pyrinomonadaceae* RB41 | yellow microbial mat from lava cave wall Portugal | 96.1 |
| 9 | AB294343 | SD4, *Pyrinomonadaceae* RB41 | stream Japan | 97.7 |
| 10 | EU132394 | SD4, *Pyrinomonadaceae* RB41 | soil from an undisturbed mixed grass prairie preserve USA | 99.2 |
| 11 | JN020220 | SD4, *Blastocatella* | Chernobyl concrete microbial biofilm Ukraine | 100.0 |
| 12 | LC026845 | SD4, *Blastocatella* | dust particles China | 96.1 |
| 13 | FJ004757 | SD2, uncultured | bulk soil Netherlands | 99.2 |
| 14 | KM200371 | SD2, uncultured | Tobacco rhizospheric soil China | 98.8 |
| 15 | FJ625349 | SD2, uncultured | boreal pine forest soil Finland | 96.5 |
| 16 | EF516082 | SD2, uncultured | grassland soil USA | 98.4 |
| 17 | EU150221 | SD2, uncultured | Soil from spruce fir forest USA | 98.0 |
| 18 | DQ450697 | SD2, uncultured | saturated alpine tundra wet meadow soil USA | 99.2 |
| 19 | KJ623626 | SD2, uncultured | volcanic ice cave sediments Antarctica | 99.6 |
| 20 | EF019283 | SD2, uncultured | trembling aspen rhizosphere USA | 98.0 |
| 21 | AB821147 | SD2, uncultured | forest soil South Korea | 98.4 |
| 22 | Y11632 | SD1, uncultured | zinc-polluted soil Belgium | 99.6 |
| 23 | HQ598413 | SD1, *Acidipila* | woodland soil Germany | 98.8 |
| 24 | JN023390 | SD1, *Granulicella* | temperate highland grassland Mexico | 95.3 |
| 25 | JN023710 | SD1, *Granulicella* | temperate highland grassland Mexico | 98.0 |
| 26 | FR667798 | SD1, *Granulicella* | iron snow from acidic coal mining-associated Lake Germany | 100.0 |
| 27 | JN023799 | SD1, *Granulicella* | temperate highland grassland Mexico | 97.6 |
| 28 | JN023530 | SD1, *Granulicella* | temperate highland grassland Mexico | 99.6 |
| 29 | JN023102 | SD1, *Bryocella* | temperate highland grassland Mexico | 98.0 |
| 30 | GU731314 | SD1, *Granulicella* | soil sample with arsenic Germany | 97.6 |
| 31 | HQ674949 | SD1, '*Acidisarcina*' | weathered feldspar mineral China | 98.8 |
| 32 | FJ625317 | SD1, *Acidipila* | boreal pine forest soil Finland | 99.6 |
| 33 | FPLS01053045 | SD1, *Granulicella* | unknown | 97.2 |
| 34 | JN023174 | SD1, *Granulicella* | temperate highland grassland | 96.9 |
| 35 | FPLL01007473 | SD1, uncultured | peat soil Japan | 100.0 |
| 36 | JN023389 | SD1, uncultured | temperate highland grassland Mexico | 96.1 |
| 37 | AB364756 | SD1, uncultured | peat soil Japan | 97.6 |
| 38 | EF018888 | *Acidobacteriales*, uncultured | trembling aspen rhizosphere USA | 96.9 |
| 39 | HM445289 | *Acidobacteriales*, uncultured | microbial mat from lava tube walls Portugal | 98.4 |
| 40 | HQ598756 | *Acidobacteriales*, uncultured | woodland soil Germany | 97.2 |
| 41 | AB364808 | *Acidobacteriales*, uncultured | peat soil Japan | 97.6 |
| 42 | FJ004744 | *Acidobacteriales*, uncultured | bulk soil Netherlands | 99.6 |
| 43 | AJ536862 | *Acidobacteriales*, uncultured | uranium mining waste pile Germany | 98.8 |
| 44 | HM445280 | *Acidobacteriales*, uncultured | microbial mat from lava tube walls Portugal | 97.2 |
| 45 | AY963371 | *Acidobacteriales*, uncultured | soil China | 98.8 |
| 46 | EF516179 | *Acidobacteriales*, uncultured | grassland soil USA | 98.4 |
| 47 | EF516150 | *Acidobacteriales*, uncultured | grassland soil USA | 97.7 |

*(Continued)*

**Table 3.** (Continued)

| № OTU | Silvadb match | Taxonomy | Reported habitat | Similarity (%) |
|---|---|---|---|---|
| 48 | JF833567 | *Acidobacteriales*, uncultured | potassium mine soil China | 98.4 |
| 49 | HM062461 | *Acidobacteriales*, uncultured | soil USA | 97.2 |
| 50 | EF516275 | *Acidobacteriales*, uncultured | grassland soil USA | 95.7 |
| 51 | HQ598546 | *Acidobacteriales*, uncultured | woodland soil Germany | 94.5 |
| 52 | EF018794 | *Acidobacteriales*, uncultured | trembling aspen rhizosphere USA | 99.2 |
| 53 | GU205282 | *Acidobacteriales*, uncultured | sediment from orthoquartzite cave Venezuela | 95.3 |
| 54 | HQ598572 | *Acidobacteriales*, uncultured | woodland soil Germany | 97.6 |
| 55 | HQ118387 | *Acidobacteriales*, uncultured | loamy soil of Eucalyptus forest USA | 98.0 |
| 56 | FJ004707 | *Acidobacteriales*, uncultured | rizosphere Lotus corniculatus Netherlands | 99.6 |
| 57 | HQ598769 | *Acidobacteriales*, uncultured | woodland soil Germany | 97.6 |
| 58 | JN023645 | *Acidobacteriales*, uncultured | temperate highland grassland Mexico | 96.9 |
| 59 | KJ410541 | *Acidobacteriales*, uncultured | Pinus massoniana soil China | 99.2 |
| 60 | FJ624925 | *Acidobacteriales*, uncultured | boreal pine forest soil Finland | 99.2 |
| 61 | EU132294 | *Holophagae*, uncultured | prairie grass soil USA | 96.5 |
| 62 | JX114379 | *Holophagae*, uncultured | rhizosphere soil Spain | 95.3 |
| 63 | EF018757 | *Holophagae*, uncultured | trembling aspen rhizosphere USA | 95.7 |
| 64 | EF516932 | *Holophagae*, uncultured | grassland soil USA | 94.9 |
| 65 | EF018864 | *Holophagae*, uncultured | trembling aspen rhizosphere USA | 98.0 |
| 66 | KJ081622 | *Holophagae*, uncultured | copper contaminated soil China | 95.7 |
| 67 | JF428950 | *Holophagae*, uncultured | rhizosphere soil China | 96.5 |
| 68 | EU132431 | *Holophagae*, uncultured | prairie grass soil USA | 96.5 |

## Discussion

All previously available knowledge of *Acidobacteria* diversity in the tundra zone refers to the ecosystems with developed plant communities, i.e. tundra heaths with a mosaic vegetation of dwarf shrubs and alpine grasses [19,34], wetlands with mixed cover of lichens and mosses [20] or forested tundra with lichen cover [35]. All of these studies report SDs 1, 2 and 3 (belonging to the class *Acidobacteriia*) as the major groups of tundra-inhabiting *Acidobacteria*. The results obtained in our study for vegetated plots (Fig 3) are in agreement with the previous reports. The search for information on *Acidobacteria* diversity in aeolian sand dunes and the corresponding chronosequence of soil formation on sands in the tundra zone, however, yielded no results. As revealed in our study, sandy soils of tundra are characterized by a prominent presence of the *Blastocatellia* (SD4), which may comprise up to one third of the acidobacterial community. This finding was somewhat unexpected because none of the currently described members of the *Blastocatellia* were obtained from cold environments. Thermophilic representatives of this class, *Pyrinomonas methylaliphatogenes* (family *Pyrinomonadaceae*) and *Chloracidobacterium thermophilum* (family '*Chloracidobacteriaceae*'), were isolated from a hot spring and a thermal soil, respectively [36,37]. All other described representatives, i.e. members of the genera *Blastocatellia*, *Aridibacter*, *Tellurimicrobium*, *Stenotrophobacter* (family *Blastocatellaceae*) and the genera *Arenimicrobium*, *Brevitalea* (family *Arenimicrobiaceae*), were obtained from Namibian semiarid savanna soils [14,38,39]. These bacteria were characterized as possessing wide growth temperature ranges (from 8–11˚C to 45–52˚C) and, yet, their growth optima were recorded at 28–45˚C. The 16S rRNA gene sequences retrieved from aeolian sands of tundra affiliated only with the families *Arenimicrobiaceae* and *Blastocatellaceae*. Most of these sequences displayed low similarity (below 91–92%) to those from described members of these families, suggesting that the phenotypes of these bacteria are also different from those of

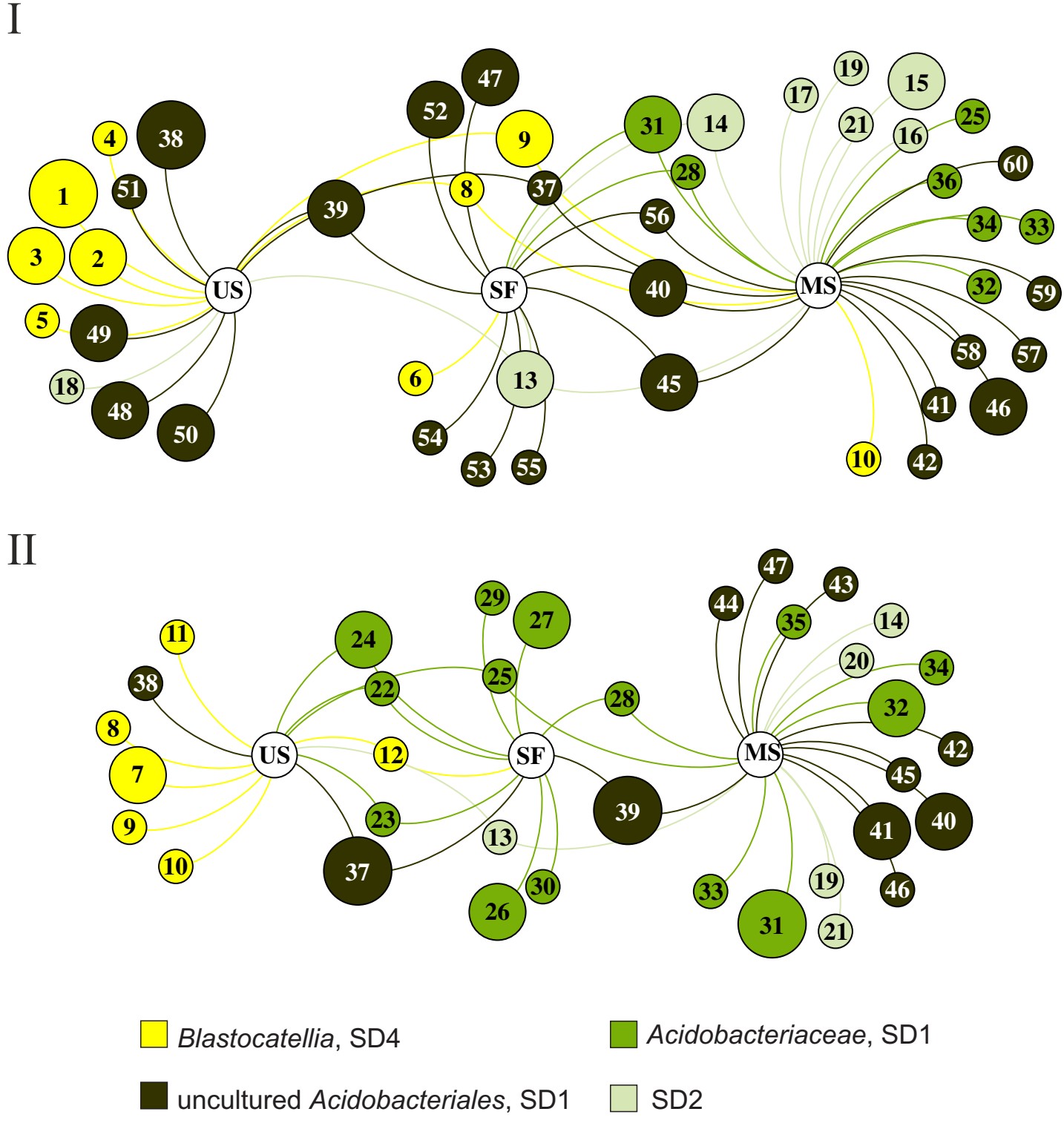

**Fig 4. Network diagram illustrating the most abundant OTU distribution between unfixed aeolian sand, semi-fixed surfaces with mosses and lichens, and mature soil under fully developed plant cover.** The size of the OTU nodes is weighted according to the relative abundance of the particular OTU. The diagram was constructed using *gephi* [29].

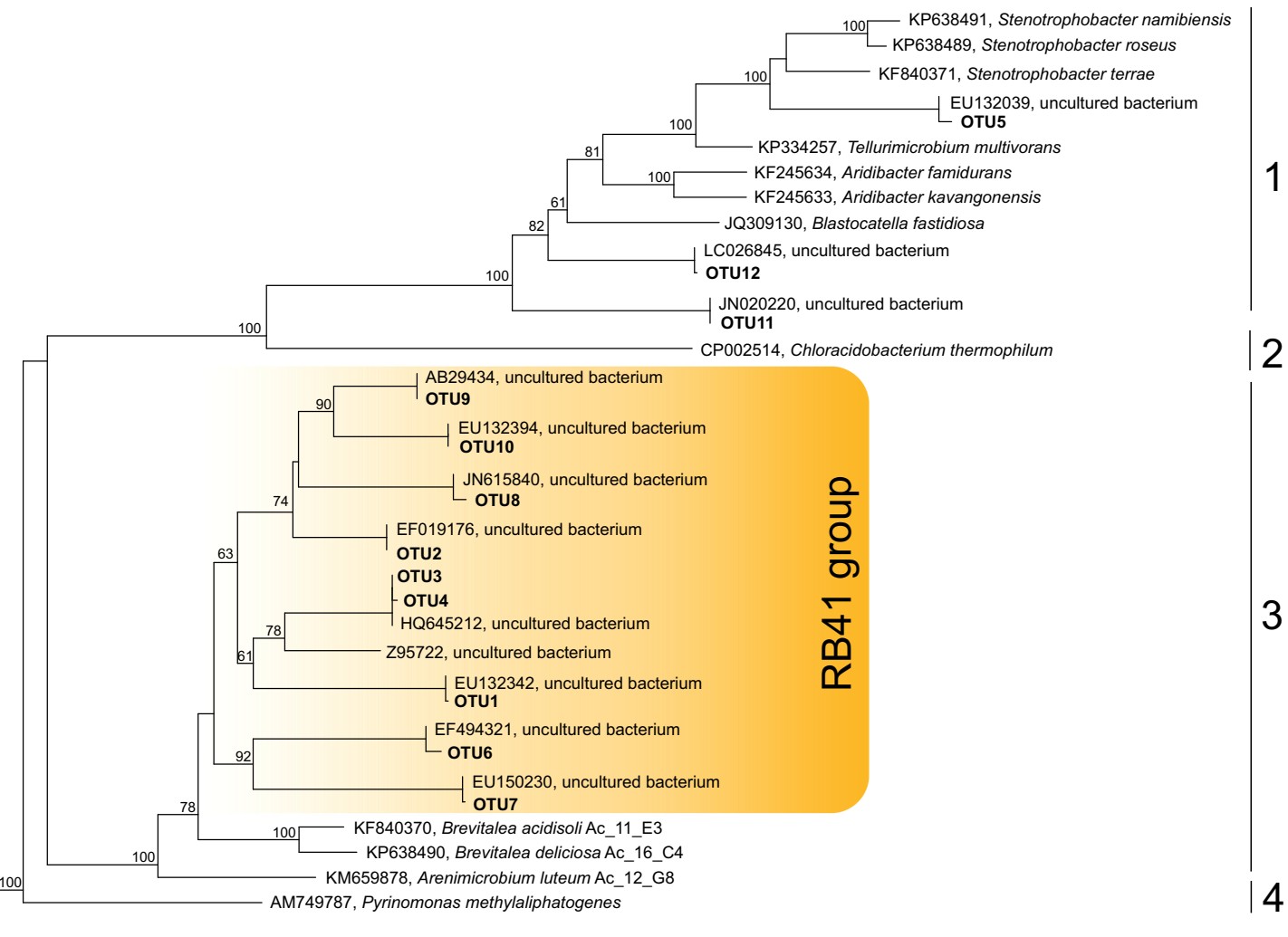

**Fig 5. Maximum parsimony tree showing the phylogenetic position of the most abundant acidobacterial OTUs from the *Blastocatellia* in relation to closest relatives of described species and/or environmental 16S rRNA gene sequences.** Various families within the *Blastocatellia* are indicated as follows: 1—*Blastocatellaceae*, 2 - '*Chloracidobacteriaceae*', 3—*Arenimicrobiaceae*, 4—*Pyrinomonadaceae*. Bootstrap values are derived from 1000 pseudoreplicates. An outgroup was composed of three 16S rRNA gene sequence from members of the *Holophagae*, *Holophaga foetida* (X77215) and two related environmental sequences (FQ658676 and FQ659446). The scale bar indicates 10% estimated sequence divergence.

acidobacteria from Namibian savanna soils. One particular group of sequences characteristic for sands of tundra, which were classified by using dbSilva 132 as 'uncultured *Pyrinomonada-ceae* RB41' (Table 3), formed a common clade with 16S rRNA gene sequences from the *Areni-microbium* and *Brevitalea* (Fig 5), suggesting that these sequences should be addressed as affiliating with the family *Arenimicrobiaceae*. Final conclusions about the taxonomic position of these bacteria, however, should await their isolation and characterization. Future efforts in culturing these bacteria could possibly benefit from the evidence for their ability to survive drought, nutrient limitation and low temperatures (as suggested by Wüst et al. [40] and our study).

The trend observed for members of the family *Acidobacteriaceae* (Table 4) agrees well with our current knowledge of their preference for low pH and high availability of plant-derived organic matter [4,6,8,10]. Notably, a large group of OTUs in semi-fixed sands or mature soils

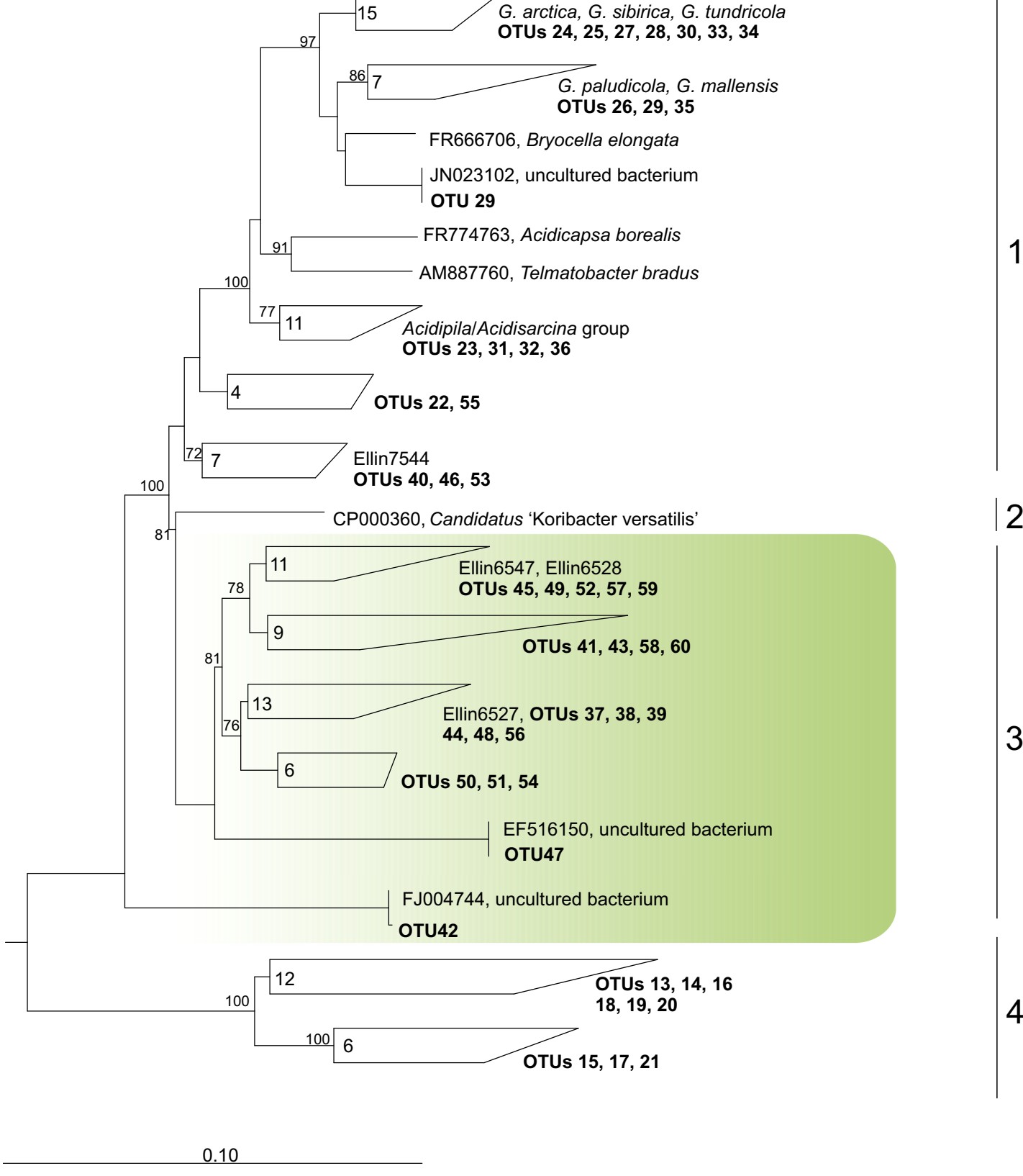

**Fig 6. Maximum parsimony tree showing the phylogenetic position of the most abundant acidobacterial sequences from the *Acidobacteriia* in relation to closest relatives of described species and/or environmental 16S rRNA gene sequences.** Various families and groups within the *Acidobacteriia* are indicated as follows: 1—*Acidobacteriaceae*, 2 –'*Koribacteraceae*', 3 –uncultured *Acidobacteriales*, 4 –SD2. Bootstrap values are derived from 1000 pseudoreplicates. An outgroup was composed of three 16S rRNA gene sequence from members of the *Holophagae*, *Holophaga foetida* (X77215) and two related environmental sequences (FQ658676 and FQ659446). The scale bar indicates 10% estimated sequence divergence.

were represented by *Granulicella* species (OTUs 24–28, 30, 33, 34). These bacteria were isolated from tundra soils or northern boreal wetlands [40–42] and are commonly associated with mosses and lichens. Good tolerance of low temperatures, wide repertoires of carbohydrate-active enzymes encoded in their genomes and pronounced hydrolytic capabilities explain wide distribution of *Granulicella* species in tundra habitats [42–45]. The OTU31, which was shared between SF and MS plots in site I and was highly abundant in MS plots of site II, is represented by '*Acidisarcina*'- like bacteria, which were also isolated from lichen-dominated tundra soils and displayed chitinolytic and xylanolytic capabilities [35].

A notable group of OTUs in unfixed sands was represented by *Holophagae*-affiliated 16S rRNA gene sequences (OTUs 61–68). This finding was somewhat unexpected because these acidobacteria are commonly not abundant in soils [9]. At present, the class *Holophagae* contains only four described representatives [12]. Three of these acidobacteria are strict anaerobes and none of them was obtained from soils. 16S rRNA gene sequences retrieved in our study from upper, aerobic layers of sandy tundra soils were only distantly related (83–86% sequence similarity) to those of described *Holophagae* members. Most likely, these sequences belong to aerobic, psychrotolerant bacteria with as-yet-unknown functional potential.

An insight into community changes of the *Acidobacteriia* may also provide valuable information regarding some as-yet-uncultivated sub-groups within this class. One example is SD2 *Acidobacteria* which, by now, does not contain characterized representatives. Similar to members of the *Acidobacteriaceae*, the relative abundance of SD2 *Acidobacteria* was negatively correlated with pH and positively correlated with organic carbon content (Table 4). However, their correlation with nitrogen availability was more strongly pronounced than that of *Acidobacteriaceae*, thus suggesting the need for increasing nitrogen content in cultivation media. Another numerically abundant group of *Acidobacteriia*-affiliated 16S rRNA gene sequences, which were retrieved from all examined samples and addressed as 'uncultured *Acidobacteriales*', was classified as belonging to an as-yet-undescribed family (Fig 6). Phylogenetic diversity within this group is relatively wide and no clear correlation was observed with any of the examined parameters (Table 4). As noticed above, several representatives of this group (isolates named Ellin7137, 7522, 6547, 6528, 6527, 5106, 323) were earlier obtained and reported as being represented by extremely slow-growing bacteria that formed mini-colonies (with

**Table 4. Correlations between pH, organic carbon, nitrogen and the number of sequences from different groups of *Acidobacteria*.**

| Acidobacterial group | pH | C | N |
|---|---|---|---|
| *Acidobacteriaceae* (SD1) | **-0.83**[*] | 0.63 | 0.56 |
| *Acidobacteriales* uncultured (SD1) | 0.37 | 0.05 | 0.11 |
| *Bryobacterales* (SD3) | 0.02 | -0.45 | -0.42 |
| SD2 | -0.73 | 0.79 | 0.80 |
| *Blastocatellia* (SD4) | 0.80 | -0.47 | -0.45 |
| SD8 | **0.82** | -0.56 | -0.59 |
| Other | 0.64 | -0.39 | -0.49 |

[*]Statistically significant values (P-value confidence level<0.05) are indicated by bold.

diameters of 25–200 μm) after long ($>$ 12 weeks) periods of incubation [33]. It is hardly surprising that none of these isolates were characterized and taxonomically described. The occurrence of these slow-growing and presumably oligotrophic bacteria in aeolian sands of tundra appears to be logical based on their phenotype. Cultivation and characterization of these microorganisms represents a challenge for further taxonomic studies on *Acidobacteria*.

## Conclusions

In summary, our analysis revealed two clearly distinct profiles of 'ecological fitness' of the *Acidobacteriia* and *Blastocatellia*. The latter, in particular members of the family *Arenimicrobiaceae*, appear to be characteristic for dry, depleted in organic matter sandy soils. An opposite habitat preference was demonstrated by the *Acidobacteriia*, which seem to specialize in degrading plant-derived organic matter and, therefore, become a major group of acidobacteria in soils under fully developed plant cover. This linkage between the taxonomic affiliation and the potential functional capabilities could help interpreting the results of molecular diversity surveys and navigate our efforts in obtaining representatives of as-yet-uncultivated groups of soil *Acidobacteria*.

## Supporting information

**S1 Table. Relative abundance of different acidobacterial groups.**
(PDF)

**S2 Table. Statistical values calculated by multiple t-test comparisons for acidobacterial groups from Sites 1 and 2.**
(PDF)

**S1 Fig. Venn diagrams showing the number of shared and unique OTUs in Sites 1 and 2, and in three types of successional stages examined in this study.**
(PDF)

**S2 Fig. Heat map showing the relative abundances of well-represented acidobacterial OTUs.**
(PDF)

## Author Contributions

**Conceptualization:** Svetlana N. Dedysh.

**Data curation:** Alena D. Zhelezova, Timofey I. Chernov.

**Formal analysis:** Anastasia A. Ivanova.

**Funding acquisition:** Timofey I. Chernov, Svetlana N. Dedysh.

**Investigation:** Alena D. Zhelezova.

**Methodology:** Anastasia A. Ivanova.

**Resources:** Alena D. Zhelezova, Timofey I. Chernov.

**Software:** Anastasia A. Ivanova, Timofey I. Chernov.

**Supervision:** Svetlana N. Dedysh.

**Visualization:** Anastasia A. Ivanova.

**Writing – original draft:** Svetlana N. Dedysh.

**Writing – review & editing:** Timofey I. Chernov, Svetlana N. Dedysh.

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
