## [Decision Letter · Decision Letter 0]

19 Dec 2019

PONE-D-19-23462

Linking ecology and systematics of acidobacteria: Distinct habitat preferences of the Acidobacteriia and Blastocatellia in tundra soils

PLOS ONE

Dear Dr. Dedysh,

Thank you for submitting your manuscript to PLOS ONE. After careful consideration, we feel that it has merit but does not fully meet PLOS ONE’s publication criteria as it currently stands. Therefore, we invite you to submit a revised version of the manuscript that addresses the points raised during the review process.

We would appreciate receiving your revised manuscript by Feb 02 2020 11:59PM. To enhance the reproducibility of your results, we recommend that if applicable you deposit your laboratory protocols in protocols.io, where a protocol can be assigned its own identifier (DOI) such that it can be cited independently in the future. For instructions see: http://journals.plos.org/plosone/s/submission-guidelines#loc-laboratory-protocols

We look forward to receiving your revised manuscript.

Kind regards,

Ying Ma, Ph.D.

Academic Editor

PLOS ONE

Journal Requirements:

3. We note that you are reporting an analysis of a microarray, next-generation sequencing, or deep sequencing data set. PLOS requires that authors comply with field-specific standards for preparation, recording, and deposition of data in repositories appropriate to their field. Please upload these data to a stable, public repository (such as ArrayExpress, Gene Expression Omnibus (GEO), DNA Data Bank of Japan (DDBJ), NCBI GenBank, NCBI Sequence Read Archive, or EMBL Nucleotide Sequence Database (ENA)). In your revised cover letter, please provide the relevant accession numbers that may be used to access these data. For a full list of recommended repositories, see http://journals.plos.org/plosone/s/data-availability#loc-omics or http://journals.plos.org/plosone/s/data-availability#loc-sequencing.

Reviewers' comments:

Reviewer's Responses to Questions

**Comments to the Author**

1. Is the manuscript technically sound, and do the data support the conclusions?

Reviewer #1: Yes

Reviewer #2: Partly

2. Has the statistical analysis been performed appropriately and rigorously? 

Reviewer #1: Yes

Reviewer #2: No

3. Have the authors made all data underlying the findings in their manuscript fully available?

Reviewer #1: Yes

Reviewer #2: Yes

4. Is the manuscript presented in an intelligible fashion and written in standard English?

Reviewer #1: Yes

Reviewer #2: Yes

5. Review Comments to the Author

Reviewer #1: Linking ecology and systematics of acidobacteria: Distinct habitat preference of the Acidobacteeria and Blastocatellia in Tundra soils (PONE-D-19-23462

Reviewed by Dr. Ali Quoreshi

Comments to Author

In general, the manuscript is well-written and has scientific merit. There are few minor corrections needed to be revised. The research tried to reveal ecological preferences of traits of acidobacteria from different classes form a gradient of sandy soils of tundra. Indeed an interesting study.

In materials and methods section (lines 102-109), it is not very clear to me about the sample size, replicates. Please provide little more details about the sampling protocol, actual numbers of true replicate from each site, is there any composite samples taken? If so then please provide details.

If I don’t misunderstand, the table 3 and Figure 3 are showing very similar information. Author may consider to adding Figure 3 in the supplementary section.

The discussion is well written. I have few concerns.

Lines 277-280, author mentioned about a shift of bacterial community and nicely shown in Figure 7. Please provide some discussions about why these shifts observed.

Authors have discussed current results and compared many other studies and appropriate references were cited. I suggest authors discussing and comparing little more in microbial compositions between the current study results and results from other ecosystems/desert/boreal/rain forest.

Table 4, please add heading under the bacterial groups listed.

Author should try to provide a better quality picture of Figure 1.

Finally, please revise the conclusions. I would like to see more relevant conclusions based on the key results and what described in the abstract.

Good luck.

Reviewer #2: The manuscript nicely describes the opposite trends observed in abundance of two acidobacterial classes along two chronosequences of soil development as observed in the boreal zone.

However, I see some room for improvement in data analysis, representation and interpretation. Major issues are that:

1) The respective stages of the two chronosequences are more or less handled/discussed as the same although obvious differences exist e.g. in mature vegetation or soil moisture and also in acidobacterial community composition (e.g. looking at the Acidobacteriaceae as a whole or looking at major OTUs from site I and II, respectively). So, I think, besides the general trend - which surely is true - this should be looked at more differentially.

2) For me there seems to be a third class (Holophagae, SD8) showing the same trend as the Blastocatellia which, however, is only briefly reported. Although abundance might a bit lower, to me this still seems worthwile more detailed analysis, especially as this – to my knowledge – would be a completely new observation made for members of this group.

3) Although the authors might not be aware of that, for my feeling data representation in some points is a bit obscure to even manipulative by e.g. stressing out single values instead of giving a mean value and a standard deviation for the replicates or not showing standard deviations for mean values. A lot of analyses are only vaguely or not at all described which raises a lot of questions.

4) Moreover a side observation from which I don’t know how relevant it is: If printed in black and white only, figures are very hard to interpret as the different colors chosen end up in very similar grey tones.

For details see the following line by line comments:

l. 66: Add “ones” between “abundant” and “in soil”.

l. 70 and many other places: Be consistent in having a space between “SD” and the respective number or not.

l. 73: “in” instead of “of”?

Whole paragraph of l. 72-83: I’d add Holophagae here, as in addition to the two classes discussed they show up in relevant fractions in the data and also contain 3 described genera.

l. 114: Information should be given in the table that values represent a single measurement of an average sample. Moreover the term “average sample” (l. 110/111) needs clarification. – Is this a pooled sample of the five replicates taken and used for the molecular analysis? Or a bigger sample taken in addition?

l. 122: I’d suggest to at least give the information here on which region (V4, in this case) of the 16S rRNA gene the analysis is based on.

l.124-126: Please clarify, if it has been only two samples in total or two from each sample type, as two samples from each sample type (US, SF and MS) must have been removed according to table 2 which doesn’t come clear from the text.

l. 126: Why no subsampling based on the sample with the lowest number of reads has been done? – Please give a reason or consider to add this step to your workflow.

l. 131: Which database (and version) has been used for phylogenetic analysis? Or haven’t sequences been added to a SILVA or living tree database first? Which algorithm(s) have been used?

l. 127: Isn’t it artificial sequence variants (ASVs) what is produced by qiime2? Please check and correct, if necessary.

l. 132: Doesn’t also the network construction need further specification?

l. 132: How where the diversity indices shown in table 2 calculated?

l. 139: I’m missing how significance of changes as given in lines 168-170 have been calculated.

l. 145: For sue it’s a true statement to report a percentage of Acidobacteria of 9-31%, but it would also be worthwhile looking at means + standard deviation of al replicates coming from the same plot. Then e.g. you would see that the sample with 31% is extremely high compared to all the others and maybe should be considered as an outlier or somewhat exceptional sample that requires to search for an explanation. Same is true for Shannon index and number of OTUs for this sample which are also higher.

l. 146: Better talk of “alpha-diversity” here?

l. 146-148: I think, this comparison can only be done on means + standard deviation. Moreover it would be important to test, if differences are statistically significant. (Same is true for number of OTUs and Pielou’s eveness that – in my opinion - moreover should be addressed in the text or can be omitted from the table, if irrelevant.)

l. 164: Add “relative” before “community composition”?

l. 167: Change “upper” and “lower” panel to “panel “A” and “B”, respectively, as depicted in the figure?

l. 168: Correct “312” to “132”.

l. 168-170: Here it would be important to know, how this result has been achieved and between which sages specifically. Just from looking at figure two I’d say that also changes for Holophagaceae must be significant between US compared to SF and MS while e.g. I wouldn’t be too confident for Acidobacteriaceae in panel B …

l. 174: For my feeling this is a big difference between the sites which should also be addressed. Moreover it should be considered to give means + standard deviation of the replicates as well.

l. 180: The statement is not really true for Acidobacteriaceae at site two. I think.

l. 182: “Acidobacteria”: write either in italics or in lowercase, depending on the meaning.

l. 188-198: How many overlap? How many are unique to sites and/or developmental stages?

l. 190 ff.: I my opinion also Holophagae should be include, if there are no good reasons why this is not feasible or relevant for the study. Moreover I wonder, why Bryobacterales have been excluded, as according to l. 168-170 they showed significant differences, while there the uncultured Acidobacteriales and the SD 2 organisms didn’t pop up (which by the way is strange at least for SD 2.

l. 198: for me figure 3 isn’t a true heat map, but rather a panel organized by plot and phylogenetic affiliation, but neither including a scale for intensity nor clustering information.

l. 200: “somewhat different” is a very vague and euphemistic circumscription for the fact that there are often completely different OTUs or at least big changes in abundance from plot to plot. In my opinion this should be addressed in more detail, as it’s also an important finding that on the one hand there is this trend on higher rank level and on the other hand this variation on OTU level. Moreover additional issues with all this are that comparisons are done on different taxonomic levels from class (Blastocatelia, Holphagae) to order or family level (within Acidobacteriia) (so, somehow (comparing pears with different sorts of apples) or comparing OTUs on around species level which I’m not sure can be resolved based on the V4 region only.

l. 201/202: I’m missing the relative abundance mentioned in the table below. Moreover I’d suggest to repeat the definition of “most abundant” from the text directly with the table. Then, I also do not fully understand the order or ranking in the table. I think, it’s neither by abundance nor fully by taxonomy … Please explain and/or consider to make it more consistent, if appropriate. Finally, the habitat information collected in the “reported habitat” column is only sparsely addressed/used which is a pity, I think.

l. 204: See comment before on the term “heat map”. Moreover, as it’s only one value, I guess, this is mean values now from the three or five replicates, respectively, which should be stated and completed by giving a standard deviation also, if my assumption is correct, or further explained, if I’m wrong.

l. 240 + 256: Is it really maximum parsimony trees? I’m just asking …

l. 266-288: this is rather a description/presentation of results than a discussion and in my opinion therefore should be moved to the results section.

l. 269: Replace “dramatic” by “pronounced”?

l. 286-288: If the size of the nodes is really based on number of OTUs and not on relative abundance, I think, this is not correct and can’t be applied, as data haven’t been normalized by subsampling to an even depth.

l. 289: Replace “of” by “on”?

l.319: Remove “[“ once.

l. 321 + 351-353: In my opinion, the table showing results should be moved to the respective section.

l. 346: Change “no surprise” to “hardly surprising”?

6. PLOS authors have the option to publish the peer review history of their article (what does this mean?). If published, this will include your full peer review and any attached files.

Reviewer #1: No

Reviewer #2: No

---

## [Author Response · Author response to Decision Letter 0]

6 Feb 2020

Reviewer #1: 

Comment: In materials and methods section (lines 102-109), it is not very clear to me about the sample size, replicates. Please provide little more details about the sampling protocol, actual numbers of true replicate from each site, is there any composite samples taken? If so then please provide details.

Response: We now provide more details regarding the sampling protocol. For molecular analyses, five soil samples-replicates (50 g each) were taken from each sampling plot and processed separately. The pooled sample of the five replicates was used only for measuring the total organic carbon and total nitrogen contents.

Comment: If I don’t misunderstand, the table 3 and Figure 3 are showing very similar information. Author may consider to adding Figure 3 in the supplementary section.

Response: This figure has been moved to the Supporting information (see Supplementary Fig S2).

Comment: Lines 277-280, author mentioned about a shift of bacterial community and nicely shown in Figure 7. Please provide some discussions about why these shifts observed.

Response: We have added one section in the Results, were we comment on correlations between chemical soil properties and the abundances of different acidobacterial groups (lines 306-316). We also discuss the reasons behind this shift in the text (lines 355-365, 376-380).

Comment: Authors have discussed current results and compared many other studies and appropriate references were cited. I suggest authors discussing and comparing little more in microbial compositions between the current study results and results from other ecosystems/desert/boreal/rain forest.

Response: We have added some information in the Introduction in order to clarify that acidobacterial SDs 1, 2, 3, 4 and 6 are the most abundant ones in soils of a wide range of ecosystem types, including boreal and tropical forests, grasslands, pastures and arid landscapes (lines 66-68). Nearly all of these studies, however, analyzed acidobacterial diversity at the level of subdivisions (SDs), which is somewhat different to the detailed taxonomic approach used in our study. 

Comment: Table 4, please add heading under the bacterial groups listed.

Response: Done.

Comment: Author should try to provide a better quality picture of Figure 1.

Response: This figure is taken from the original study of Zhelezova et al. 2019 published in PLoS ONE. We have made an attempt to improve it. It now looks fine when printed in its final size.

Comment: Finally, please revise the conclusions. I would like to see more relevant conclusions based on the key results and what described in the abstract.

Response: The conclusions have been revised.

Reviewer #2: 

Comment: Major issues are that:

1) The respective stages of the two chronosequences are more or less handled/discussed as the same although obvious differences exist e.g. in mature vegetation or soil moisture and also in acidobacterial community composition (e.g. looking at the Acidobacteriaceae as a whole or looking at major OTUs from site I and II, respectively). So, I think, besides the general trend - which surely is true - this should be looked at more differentially.

Response: The logic behind our current analysis is clear: we trace the changes in acidobacterial diversity over a gradient of increasing vegetation. The rationale behind comparing acidobacterial communities in unfixed sands of Nelmin Nos and Naryan-Mar is less clear and may not be of interest to the reader. However, we have followed this request to some extent and compared the pools of OTUs obtained from the two sites and presented the results of this analysis in Supplementary Figures S1 and S2. We have also added one text paragraph in order to address this difference (lines 232-239).

Comment: 2) For me there seems to be a third class (Holophagae, SD8) showing the same trend as the Blastocatellia which, however, is only briefly reported. Although abundance might a bit lower, to me this still seems worthwile more detailed analysis, especially as this – to my knowledge – would be a completely new observation made for members of this group.

Response: The information about Holophagae-affiliated Acidobacteria has been incorporated in the revised manuscript. In particular, the most abundant OTUs of these bacteria are now included in Table 3. We also provide some discussion about this class of Acidobacteria (lines 366-373). However, as could be seen from Supplementary Table S1, the relative abundance of these bacteria did not display statistically significant difference between different plots examined in our study. 

Comment: 3) Although the authors might not be aware of that, for my feeling data representation in some points is a bit obscure to even manipulative by e.g. stressing out single values instead of giving a mean value and a standard deviation for the replicates or not showing standard deviations for mean values. A lot of analyses are only vaguely or not at all described which raises a lot of questions.

Response: We have added all requested details of our bioinformatics and statistical analyses in the Methods (see our specific responses below). The revised manuscript version includes the file with Supplementary materials, which has been prepared in order to address all concerns of the reviewer.

Comment: 4) If printed in black and white only, figures are very hard to interpret as the different colors chosen end up in very similar grey tones.

Response: PLoS ONE is an online journal; no printed version of this journal is available, which implies free choice of colors. The practice of printing out selected papers for personal use is also nearly gone. We hope there are no problems with our color figures.

Comment: l. 66: Add “ones” between “abundant” and “in soil”.

Response: corrected as recommended.

Comment: l. 70 and many other places: Be consistent in having a space between “SD” and the respective number or not.

Response: We have chosen not to have a space between “SD” and the respective number. A space is used only if several SDs are mentioned (for example SDs 1 and 3).

Comment: l. 73: “in” instead of “of”?

Response: Ok, done.

Comment: Whole paragraph of l. 72-83: I’d add Holophagae here, as in addition to the two classes discussed they show up in relevant fractions in the data and also contain 3 described genera.

Response: This is not a proper place to discuss Holophagae since this class, in contrast to Acidobacteriia and Blastocatellia, contains very few described genera. However, we now comment on the Holophagae in the Results and Discussion (see specific comments below). 

Comment: l. 114: Information should be given in the table that values represent a single measurement of an average sample. Moreover the term “average sample” (l. 110/111) needs clarification. – Is this a pooled sample of the five replicates taken and used for the molecular analysis? Or a bigger sample taken in addition?

Response: Our description was not fully correct, we agree. The term “average sample” has been replaced with “a pooled sample of the five replicates taken for the molecular analysis”. We also clarify in Table 1 that values are shown as means (n=2 for TOC and TN, n=5 for pH) ± standard deviations).

Comment: l. 122: I’d suggest to at least give the information here on which region (V4, in this case) of the 16S rRNA gene the analysis is based on.

Response: Ok, we have included this info in the revised manuscript.

Comment: l.124-126: Please clarify, if it has been only two samples in total or two from each sample type, as two samples from each sample type (US, SF and MS) must have been removed according to table 2 which doesn’t come clear from the text.

Response: These were only two replicates of MS plot in site I. We clarify this now in the revised manuscript.

Comment: l. 126: Why no subsampling based on the sample with the lowest number of reads has been done? – Please give a reason or consider to add this step to your workflow.

Response: Unfortunately, this was no longer possible. The sampling was performed in August 2015, while our study was initiated in 2019. All collected soil samples and DNA extracts were fully utilized by that time.

Comment: l. 131: Which database (and version) has been used for phylogenetic analysis? Or haven’t sequences been added to a SILVA or living tree database first? Which algorithm(s) have been used?

Response: We used the ARB program package (version 6.0.3) and the sequence alignment based on dbSilva132. Short sequences were added to the existing tree using the quick ARB parsimony insertion tool. These details are now provided in the revised manuscript.

Comment: l. 127: Isn’t it artificial sequence variants (ASVs) what is produced by qiime2? Please check and correct, if necessary.

Response: No, our text is correct. In QIIME 2, all sequences before clustering with certain identity level are addressed as amplicon sequence variants (ASVs). The use of VSEARCH plugin results in obtaining operational taxonomic units.

Comment: l. 132: Doesn’t also the network construction need further specification?

Response: Gephi is a program with a simple Excel-like interface. It uses two spreadsheets as an input (for nodes and edges, respectively), so that no specific parameters could be added to our description.

Comment: l. 132: How where the diversity indices shown in table 2 calculated?

Response: These indices were calculated using the core-metrics-phylogenetic method implemented in in QIIME 2 v.2018.8. We clarify this in the revised manuscript.

Comment: l. 139: I’m missing how significance of changes as given in lines 168-170 have been calculated.

Response: These values have been calculated by multiple t-test comparisons. We have added some details in the Methods. The corresponding table with these values is now included in the Supporting information (Table S2). 

Comment: l. 145: For sue it’s a true statement to report a percentage of Acidobacteria of 9-31%, but it would also be worthwhile looking at means + standard deviation of al replicates coming from the same plot. Then e.g. you would see that the sample with 31% is extremely high compared to all the others and maybe should be considered as an outlier or somewhat exceptional sample that requires to search for an explanation. Same is true for Shannon index and number of OTUs for this sample which are also higher.

Response: We do not make an attempt to compare the percentage of Acidobacteria in different plots. Most readers are interested to see the range (9-31%). However, since we do compare the acidobacterial alpha-diversity diversity in the text (lines 160-162), we have followed reviewer’s advice and expressed Shannon indexes in Table 2 as means + standard deviation. We also state now that the acidobacterial alpha-diversity unfixed aeolian sands was slightly lower than that in soils of vegetated plots but this difference was not statistically significant.

Comment: l. 146: Better talk of “alpha-diversity” here?

Response: corrected.

Comment: l. 146-148: I think, this comparison can only be done on means + standard deviation. Moreover it would be important to test, if differences are statistically significant. (Same is true for number of OTUs and Pielou’s eveness that – in my opinion - moreover should be addressed in the text or can be omitted from the table, if irrelevant.)

Response: We agree. The individual values of Shannon indexes in Table 2 have been replaced with the corresponding mean values. The differences in acidobacterial diversity between different plots were not statistically significant. We clarify this now in the text.

Comment: l. 164: Add “relative” before “community composition”?

Response: Sorry, this is not a good advice. The wording “relative community composition” has no sense. It is clear from the figure caption that this diagram was constructed based on a relative abundance of particular acidobacterial groups.

Comment: l. 167: Change “upper” and “lower” panel to “panel “A” and “B”, respectively, as depicted in the figure?

Response: Done.

Comment: l. 168: Correct “312” to “132”.

Response: Corrected.

Comment: l. 168-170: Here it would be important to know, how this result has been achieved and between which sages specifically. Just from looking at figure two I’d say that also changes for Holophagaceae must be significant between US compared to SF and MS while e.g. I wouldn’t be too confident for Acidobacteriaceae in panel B …

Response: The table containing the corresponding set of statistical values is now included in the Supplementary materials (Table S1). As seen from this table, no significant changes were detected for the Holophagaceae. Significant changes, however, were revealed for the Acidobacteriaceae in US/SF and US/SM samples from both sites.

Comment: l. 174: For my feeling this is a big difference between the sites which should also be addressed. Moreover it should be considered to give means + standard deviation of the replicates as well.

Response: We now provide these values as means + standard deviations of the replicates in Supplementary Table S1 and we also replaced ranges with means + standard deviations in the text.

Comment: l. 180: The statement is not really true for Acidobacteriaceae at site two. I think.

Response: We have revised this statement in order to separately address the relative abundances of Acidobacteriaceae (SD1) in US plots of sites I and II.

Comment: l. 182: “Acidobacteria”: write either in italics or in lowercase, depending on the meaning.

Response: corrected.

Comment: l. 188-198: How many overlap? How many are unique to sites and/or developmental stages?

Response: The requested information is now given in the text and represented as a Venn diagram in Supplementary Fig. S1.

Comment: l. 190 ff.: I my opinion also Holophagae should be include, if there are no good reasons why this is not feasible or relevant for the study. Moreover I wonder, why Bryobacterales have been excluded, as according to l. 168-170 they showed significant differences, while there the uncultured Acidobacteriales and the SD 2 organisms didn’t pop up (which by the way is strange at least for SD 2.

Response: We have followed reviewer’s request and included some data for Holophagae as well (see Table 3 and Supplementary Fig S2). We have also included some discussion on this group of Acidobacteria (Discussion, lines 366-373). However, we would like to keep our major focus on Acidbacteriia and Blastocatellia (as specified in the manuscript title). Focusing on all groups of Acidobacteria would make the whole story too complex and “heavy”.

Comment: l. 198: for me figure 3 isn’t a true heat map, but rather a panel organized by plot and phylogenetic affiliation, but neither including a scale for intensity nor clustering information.

Response: As recommended by Referee1, this figure has been moved to the Supplementary materials (see Figure S2). We also included an intensity scale in this figure.

Comment: l. 200: “somewhat different” is a very vague and euphemistic circumscription for the fact that there are often completely different OTUs or at least big changes in abundance from plot to plot. In my opinion this should be addressed in more detail, as it’s also an important finding that on the one hand there is this trend on higher rank level and on the other hand this variation on OTU level. Moreover additional issues with all this are that comparisons are done on different taxonomic levels from class (Blastocatelia, Holphagae) to order or family level (within Acidobacteriia) (so, somehow (comparing pears with different sorts of apples) or comparing OTUs on around species level which I’m not sure can be resolved based on the V4 region only.

Response: The referee is right. Diversity of OTUs identified in the sites I and II was clearly different, which is nicely illustrated by Supplementary Fig S2.We have added one text paragraph in order to address this difference (lines 232-239).

Comment: l. 201/202: I’m missing the relative abundance mentioned in the table below. Moreover I’d suggest to repeat the definition of “most abundant” from the text directly with the table. Then, I also do not fully understand the order or ranking in the table. I think, it’s neither by abundance nor fully by taxonomy … Please explain and/or consider to make it more consistent, if appropriate. Finally, the habitat information collected in the “reported habitat” column is only sparsely addressed/used which is a pity, I think.

Response: This is our mistake; we apologize. The relative abundance was not included in this table. This information is now provided in Supplementary Figure S2, because the corresponding values differ for the two sites. The OTUs are arranged according to their numbers generated by QIIME. It is simply impossible to arrange them by abundance because it differs between the sites (one particular OTU may be highly abundant in one site and totally absent from the second site).

Comment: l. 204: See comment before on the term “heat map”. Moreover, as it’s only one value, I guess, this is mean values now from the three or five replicates, respectively, which should be stated and completed by giving a standard deviation also, if my assumption is correct, or further explained, if I’m wrong.

Response: This figure has been corrected and moved to the Supporting information (see our replay above).

Comment: l. 240 + 256: Is it really maximum parsimony trees? I’m just asking …

Response: Yes, these are maximum parsimony trees with OTUs added via quick arb parsimony insertion tool.

Comment: l. 266-288: this is rather a description/presentation of results than a discussion and in my opinion therefore should be moved to the results section.

Response: We agree. These text fragments and the corresponding figures have been moved to the Results.

Comment: l. 269: Replace “dramatic” by “pronounced”?

Response: Done.

Comment: l. 286-288: If the size of the nodes is really based on number of OTUs and not on relative abundance, I think, this is not correct and can’t be applied, as data haven’t been normalized by subsampling to an even depth.

Response: We thank the referee for pointing out to this mistake in the figure caption. Of course, the size of the nodes was weighted according to the relative abundance of particular OTUs. The figure caption has been corrected.

Comment: l. 289: Replace “of” by “on”?

Response: According to the results of our search in the literature, this does not look like a good advice.

Comment: l.319: Remove “[“ once.

Response: Done.

Comment: l. 321 + 351-353: In my opinion, the table showing results should be moved to the respective section.

Response: Table 4 and the corresponding discussion have been moved to the Results.

Comment: l. 346: Change “no surprise” to “hardly surprising”?

Response: Done.

---

## [Decision Letter · Decision Letter 1]

24 Feb 2020

Linking ecology and systematics of acidobacteria: Distinct habitat preferences of the Acidobacteriia and Blastocatellia in tundra soils

PONE-D-19-23462R1

Dear Dr. Dedysh,

We are pleased to inform you that your manuscript has been judged scientifically suitable for publication and will be formally accepted for publication once it complies with all outstanding technical requirements.

With kind regards,

Ying Ma, Ph.D.

Academic Editor

PLOS ONE

Additional Editor Comments (optional):

Reviewers' comments:

Reviewer's Responses to Questions

**Comments to the Author**

1. If the authors have adequately addressed your comments raised in a previous round of review and you feel that this manuscript is now acceptable for publication, you may indicate that here to bypass the “Comments to the Author” section, enter your conflict of interest statement in the “Confidential to Editor” section, and submit your "Accept" recommendation.

Reviewer #2: (No Response)

2. Is the manuscript technically sound, and do the data support the conclusions?

Reviewer #2: (No Response)

3. Has the statistical analysis been performed appropriately and rigorously? 

Reviewer #2: (No Response)

4. Have the authors made all data underlying the findings in their manuscript fully available?

Reviewer #2: (No Response)

5. Is the manuscript presented in an intelligible fashion and written in standard English?

Reviewer #2: (No Response)

6. Review Comments to the Author

Reviewer #2: (No Response)

7. PLOS authors have the option to publish the peer review history of their article (what does this mean?). If published, this will include your full peer review and any attached files.

Reviewer #2: No

---

## [Editor Report · Acceptance letter]

6 Mar 2020

PONE-D-19-23462R1 

Linking ecology and systematics of acidobacteria: Distinct habitat preferences of the *Acidobacteriia* and *Blastocatellia* in tundra soils 

Dear Dr. Dedysh:

I am pleased to inform you that your manuscript has been deemed suitable for publication in PLOS ONE. Congratulations! Your manuscript is now with our production department. 

With kind regards,

on behalf of

Dr. Ying Ma 

Academic Editor

PLOS ONE